# Machine learning for healthcare that matters: Reorienting from technical novelty to equitable impact

**Aparna Balagopalan**[1], **Ioana Baldini**[2], **Leo Anthony Celi**[3,4,5], **Judy Gichoya**[6], **Liam G. McCoy**[7]*, **Tristan Naumann**[8], **Uri Shalit**[9], **Mihaela van der Schaar**[10,11], **Kiri L. Wagstaff**[12]

1 Department of Electrical Engineering and Computer Science, Massachusetts Institute of Technology; Cambridge, Massachusetts, United States of America, 2 IBM Research; Yorktown Heights, New York, United States of America, 3 Laboratory for Computational Physiology, Massachusetts Institute of Technology; Cambridge, Massachusetts, United States of America, 4 Division of Pulmonary, Critical Care and Sleep Medicine, Beth Israel Deaconess Medical Center; Boston, Massachusetts, United States of America, 5 Department of Biostatistics, Harvard T.H. Chan School of Public Health; Boston, Massachusetts, United States of America, 6 Department of Radiology and Imaging Sciences, School of Medicine, Emory University; Atlanta, Georgia, United States of America, 7 Division of Neurology, Department of Medicine, University of Alberta; Edmonton, Alberta, Canada, 8 Microsoft Research; Redmond, Washington, United States of America, 9 The Faculty of Data and Decision Sciences, Technion; Haifa, Israel, 10 Department of Applied Mathematics and Theoretical Physics, University of Cambridge; Cambridge, United Kingdom, 11 The Alan Turing Institute; London, United Kingdom, 12 Independent Researcher; United States of America

☯ These authors contributed equally to this work.
* lmccoy@ualberta.ca

**Data Availability Statement:** All data is contained within the manuscript.

**Funding:** A.B. was funded in part by an Amazon Science PhD Fellowship at the MIT Science Hub. I.

## Abstract

Despite significant technical advances in machine learning (ML) over the past several years, the tangible impact of this technology in healthcare has been limited. This is due not only to the particular complexities of healthcare, but also due to structural issues in the machine learning for healthcare (MLHC) community which broadly reward technical novelty over tangible, equitable impact. We structure our work as a healthcare-focused echo of the 2012 paper "Machine Learning that Matters", which highlighted such structural issues in the ML community at large, and offered a series of clearly defined "Impact Challenges" to which the field should orient itself. Drawing on the expertise of a diverse and international group of authors, we engage in a narrative review and examine issues in the research background environment, training processes, evaluation metrics, and deployment protocols which act to limit the real-world applicability of MLHC. Broadly, we seek to distinguish between *machine learning ON healthcare data* and *machine learning FOR healthcare*—the former of which sees healthcare as merely a source of interesting technical challenges, and the latter of which regards ML as a tool in service of meeting tangible clinical needs. We offer specific recommendations for a series of stakeholders in the field, from ML researchers and clinicians, to the institutions in which they work, and the governments which regulate their data access.

B. is an employee of IBM Research. J.G. is a 2022
Robert Wood Johnson Foundation Harold Amos
Medical Faculty Development Program and
declares support from RSNA Health Disparities
grant (#EIHD2204), Lacuna Fund (#67), Gordon
and Betty Moore Foundation, and NIH (NIBIB)
MIDRC grant under contracts 75N92020C00008
and 75N92020C00021. U.S. is partially funded by
the Israeli Science Foundation grant no. 1950/19.
L.A.C. is funded by the National Institute of Health
through NIBIB R01 EB017205. T.N. is an employee
of Microsoft Research. M.vdS. is supported by the
Office of Naval Research (ONR). The funders had
no role in study design, data collection and
analysis, decision to publish, or preparation of the
manuscript.

**Competing interests:** L.A.C. is the Editor-in Chief
of PLOS Digital Health.

### Author summary

The field of machine learning has made significant technical advancements over the past
several years, but the impact of this technology on healthcare practice has remained lim-
ited. We identify issues in the structure of the field of machine learning for healthcare
which incentivise work that is scientifically novel over work that ultimately impacts
patients. Among others, these issues include a lack of diversity in available data, an
emphasis on targets which are easy to measure but may not be clinically important, and
limited funding for work focused on deployment. We offer a series of suggestions about
how best to address these issues, and advocate for a distinction to be made between
"machine research performed ON healthcare data" and true "machine FOR healthcare".
The latter, we argue, requires starting from the very beginning with a focus on the impact
that a model will have on patients. We conclude with discussion of "impact challenges"—
specific and measurable goals with an emphasis upon health equity and broad community
impact—as examples of the types of goals the field should strive toward.

## Introduction

The 2012 paper "Machine Learning that Matters" [1] introduced Impact Challenges as a mecha-
nism for opening discussion in the machine learning (ML) community about the limitations in
datasets, metrics, and the impact of results in the context of their originating domains. The arti-
cle presents the ML community as overly concerned with theoretical, benchmark-focused work
detached from real-world problems, and ultimately failing to address tangible issues. Despite
the incredible progress that has been made in ML [2], particularly in the field of large generative
models [3], many of the paper's original criticisms continue to ring true. We contend that these
issues are of particular concern in the field of machine learning for healthcare (MLHC).

Despite its challenges, such as the safety-critical [4] nature of decisions and the complex
multi-stakeholder [5] environments, healthcare is in many ways an ideal setting for the appli-
cation of machine learning [6]. The amount of data involved is vast [7], and the setting is
deeply meaningful. Research in this space can have immense value [6]–from providing timely,
more precise and more objective decision support for providers and patients [8,9], to cognitive
offloading for overburdened healthcare workers [10,11], to improving health system efficiency
and population health outcomes [12].

Yet, in our opinion, there exists a stark contrast between the stated ambition (and associ-
ated hype) of the MLHC field and its limited degree of meaningful impact to date. The
approach of the field to the COVID-19 pandemic serves as an illustrative example. With the
pandemic affecting countries world-wide, there were multiple coordinated efforts to collect
and share data for ML-based COVID-19 diagnosis [13]. However, despite the generation of
several hundred published predictive models, systematic reviews found severe roadblocks to
the use of these models in clinical practice [14,15]. Particularly, only a handful were promising
enough for deployment in real clinical settings [16].

We believe that this example, while an extreme case, is not an outlier. While technical tools
in the space of ML for healthcare have made great leaps [17], social aspects of such tools in
practice–be it their design, development or deployment–are often afterthoughts [18–20]. Mod-
els are too often developed without appropriate scrutiny of underlying data and subsequent
outputs [14,21]. Therefore, machine learning models learn and ultimately reproduce the biases
of society [22] at large, as reflected in racial differences in care patterns [23–27], poor

**Framework**

| Environment: Background for Building | Process: Training for Impact | Evaluation: Re-Aligning Incentives | Deployment: From Code to Clinic |
|---|---|---|---|
| 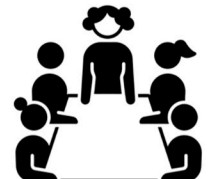 | 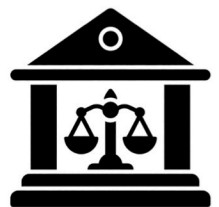 | 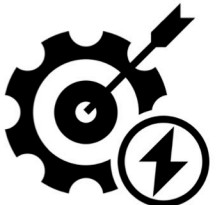 | 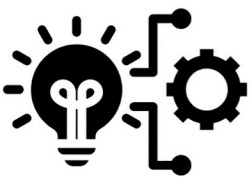 |

**Recommendations**

- ✦ Understanding social determinants & data biases
- ✦ Improving diversity among researchers
- ✦ Fostering diverse data creation
- ✦ Opening data & increasing accessibility
- ✦ Building foundational tools for MLHC success

- ✦ Selecting meaningful tasks
- ✦ Moving past the status quo
- ✦ Establishing consistent responsibility standards
- ✦ Operationalizing fairness

- ✦ Recognizing advances for more than just technical novelty
- ✦ Moving on from clinically unimportant metrics
- ✦ Encouraging moonshots

- ✦ Valuing engineering effort
- ✦ Considering human processes
- ✦ Identifying system limitations
- ✦ Building for continued impact
- ✦ Valuing generalization on par with innovation

**Fig 1. Machine learning for healthcare impact framework and recommendations.**

performance of clinical devices such as oxygen saturation probes for patients with dark skin [28,29], and numerous other aspects of the broader care system [30].

The MLHC field has, in our opinion, failed to address these issues, and has indeed exacerbated them through structures which broadly reward technically novel innovations over tangible clinical impacts. Throughout this paper, we will explore issues in the background environment, development processes, research evaluation, and clinical deployment of MLHC projects. We seek to critically analyze misaligned incentives which have led to a stark disconnect between our field's research and its broader societal impact. While several prior works [6,31–37] have addressed this topic to varying degrees, we intend this work to summarize gaps and pain points, and we present recommendations to re-center the focus of machine learning as applied to healthcare.

We conclude with a series of renewed impact challenges, hoping to re-center the notion that the core purpose of MLHC is to achieve equitable impact on meaningful clinical outcomes. All aspects of MLHC—background environment, development processes, evaluation metrics, and deployment efforts–must be understood in terms of how they advance or hinder such tangible goals. An overview of our suggestions can be seen in Fig 1.

## Environment: Background for building

The context in which research is conducted serves as the foundational platform, influencing the formation, development, and impact of research findings. Particularly in the rapidly-evolving field of MLHC, the broader research environment sets the stage for how and what kind of knowledge is generated. It is crucial to recognize that the nature and quality of data, the accessibility of this data, the inherent biases present, and the diversity of those conducting research all converge to form this dynamic milieu. The importance of this research background extends beyond the realm of technical advancements, shaping the ethical, societal, and practical implications of AI technologies.

## Understanding social determinants and data biases

Disparities in health and healthcare are present along numerous axes, including but far from limited to race, gender, socioeconomic status, and nationality [27]. Uncritically trained MLHC models tend to learn, reproduce, and further entrench these disparities—as shown in the prominent case of an algorithm which used past spending received as a proxy for health needs and further disadvantaged Black patients [27]. It is not enough to simply remove these attributes from the datasets, as ML models have demonstrated a concerning degree of ability to ascertain protected characteristics even in the absence of obvious proxies [38]. The data produced by social processes cannot be understood in isolation from those processes [39]. Similarly, work must be done not merely to eliminate the biases in health data, but to actively characterize and counteract the conditions which generate these biases in the first place. Researchers must seek to achieve "data empathy" [40], and understand the intensely human processes and stories that data—particularly data in healthcare—embed [41].

We recommend:

- Work to actively characterize and counteract underlying disparities and social determinants of health embedded in healthcare data, particularly in international contexts.

- Engage critically with processes underlying data production, and seek both upstream and downstream measures to ensure that all patients are equitably represented and impacted.

- Ensure that patient perspectives and insights are represented during data creation, validation, and release phases.

- Pursue the goal of "data empathy", ensuring that the human stories underlying medical data are not forgotten amidst the abstracting processes of MLHC.

## Improving diversity among researchers

Illness impacts us all, but the burdens of both disease and substandard care tend to fall most heavily upon those who are already marginalized [42]. However, neither the demography nor the processes of the MLHC field reflect this reality. Both the broader MLHC field at large and the subfield of fairness research are disproportionately composed of male, white, and Asian researchers from a small set of high-income countries [43]. The research community as a whole also skews both younger and more able bodied than broader patient populations [44]. The result is a field with significant blind spots, and a broader failure to adequately consider and align with the voices of those who have greatest need [45]. In order to truly achieve impact, the MLHC field must embrace diversity in its deepest sense, with researchers from a wide range of racial, gender, and national backgrounds.

The need for diversity also includes the training background and professional roles of those involved, and a wide range of stakeholders must be actively invited in at every stage of the research process. In this regard, the MLHC field should engage with and build upon existing "patient engagement" practices in clinical trials [46], which seek to ensure that research is aligned with patient needs [47–49]. With MLHC envisioned to impact a broad range of healthcare contexts, all of those involved may have valuable insights in co-designing projects [50]—including patients, caregivers, and practitioners historically underrepresented in the academic research process such as homecare workers or nursing aides [51].

We recommend:

- Promote MLHC training to a wide range of health and computer science trainees, with a particular emphasis upon members of underrepresented groups.

- Carry out international events such as datathons [52] to build global collaborative networks, ensuring that the agenda of the field is not controlled by those in a narrow set of high-income countries.

- Actively invite patient and caregiver stakeholders into all levels of the process, ensuring that procedures and outputs are aligned with tangible patient needs.

- Emphasize diversity in training and practice contexts (such as inclusion of clinicians from minority-serving institutions in high income countries), in addition to diversity in background.

- Improve the upstream training pipeline through investment in developing foundational data science capacity at minority-serving institutions such as historically Black colleges and universities,

## Fostering diverse dataset creation

Machine learning thrives on data, yet this is not fully reflected in current conference proceedings, where papers that discuss the development of datasets are rare. Some venues have emerged (such as the NeurIPS Datasets and Benchmarks track) [53], but they remain relatively novel and rare. The causes of this are multifactorial. Developing datasets requires significant effort, particularly in complex domains such as healthcare [54], and requires collaborations [55], resources, and funding which are often limited to only the largest of institutions [56]. Concerns regarding patient privacy and downstream data use further limit these collaborations specifically in the healthcare context.

As a consequence, a significant proportion of machine learning work tends to be excessively focused on a few publicly available datasets [57]. When these datasets are not sufficient for data hungry models, researchers can combine multiple datasets resulting in the so-called Frankenstein datasets [14], which are difficult to audit and can have misleading assumptions when a class label is assigned. ML works related to healthcare are subsequently limited in scope to a small set of problems such as prediction of mortality [58] or hospital length of stay [59], while widespread issues of critical importance such as maternal mortality [60,61] or discrepancies in healthcare access remain underexplored [62]. Further, these few datasets are skewed toward representing the subset of patients served by large institutions in affluent geographic locations [63]. Subsequently, algorithms developed using these datasets may serve to further exacerbate disparities to the disadvantage of underrepresented patient populations.

There are a growing range of initiatives designed to increase the diversity of dataset creation. NIH projects such as AIM-AHEAD [64] and BRIDGE2AI [65] aim to expand health data science capacity throughout the United States. Capacity in developing countries is being fostered by programs such as the NIH DS-I Africa program [66], as well as initiatives from the Gates Foundation [67] and Wellcome Trust [68]. It is critical to ensure that these capacity building efforts are not merely limited to the provision of funding, but also include work to foster a data sharing culture, address local privacy concerns [69,70], and promote multidisciplinary collaboration [71]. Local leadership and agency in this regard must remain paramount, in order to avoid perpetuation of global health power imbalances. In addition, datasets from underrepresented sources must be actively included in the mainstream of MLHC research, so that cutting-edge core models and methodologies are representative of *all* patients.

We recommend:

- Recognize the critical importance of dataset creation work in funding, hiring, and promotion contexts.

- Focus the creation of novel datasets on rectifying existing issues of patient underrepresentation.

- Provide detailed dataset descriptions (such as those described in "Datasheets for Datasets" [72,73]) that inform data harmonization practices, especially when the data use is different from its original intended use.

- Ensure that datasets created in low resource settings are included in the mainstream advancement of MLHC, rather than being limited to regional use.

## Opening data and increasing accessibility

Where robust and comprehensive datasets do exist, they are often subject to significant restrictions which limit their accessibility and usefulness to the MLHC community [57]. This may be due to concerns from host health organizations regarding patient privacy or re-identification risk [74,75]. There may also be a desire on the part of dataset creators to maintain exclusivity and reap the professional rewards of their hard work in creating the dataset in the first place.

Experience with existing open datasets, such as the Medical Information Mart for Intensive Care (MIMIC) [76], has demonstrated both sets of concerns to be often overblown [77]. With respect to privacy, de-identification measures are highly effective in practical terms, and there have been no known instances of re-identification of individuals in this dataset since its initial release in 1996 [78]. With respect to research output, experience has shown that open datasets beget a synergistic momentum, with multiple groups able to both collaborate and compete to drive the field forward. As the 5500+ citations of the MIMIC-III database have demonstrated [76], there is no shortage of clinical questions to be answered on a given set of data.

We recommend:

- Promote the creation of FAIR (Findable, Accessible, Interoperable, and Reusable) [79] datasets in healthcare, alongside the sharing of relevant supportive resources and code for dataset cleaning and optimization.

- Expand privacy-preserving efforts in areas such as vectorization, synthetic data, and federated learning, while remaining wary of their limitations and favoring completely open data sharing where possible.

- Ensure that efforts toward open data are inclusive of a broad range of healthcare settings, both internationally and within a given country. This includes not only large academic tertiary hospitals, but also regional hospitals, and alternative care settings such as community clinics.

## Building foundational tools for MLHC success

If the benefits of machine learning in healthcare are to be realized equitably, they must be realized at scale within numerous diverse clinical contexts. Where this localization does occur, it is often performed in labor-intensive ways by high-cost teams of experts, who often find themselves reinventing the proverbial wheel. For scalable success, MLHC must embrace the spirit of the open source movement, and work collaboratively to develop open automated methods to assist in model development, localization, and validation [80]. A notable and successful effort in this direction is the work done by the OHDSI consortium [81]. The field must recognize and reward such infrastructural work, given its critical importance to achieving scalable impact.

We recommend:

- Develop and standardize ML methods for data harmonization, reducing the significant existing barriers to bringing together data from multiple clinical sources.

- Require sharing of data cleaning and preprocessing code, in addition to final model development code, with detailed performance breakdowns for data subgroups using reporting tools like model cards [82].

- Create automated systems for comparing methods and facilitating evaluation.

- Create open-sourced autoML frameworks to automatically compare developed models systematically on important metrics. [80]

- Contribute to developing robust open frameworks for enduring research data sharing and distribution (such as dataverse). [80]

## Process: Training for impact

In the ever-evolving landscape of machine learning in healthcare (MLHC), the importance of process—how we frame, conduct, and evaluate our research—is paramount. It's not merely a question of pursuing advanced technology as abstractly defined. Rather the process itself holds the key to ensuring the relevance, applicability, and ethical integrity of our findings. The forthcoming section explores this fundamental concept, illustrating how meaningful task selection, ambitious benchmarking, consistent responsibility standards, and operationalized fairness are critical in shaping the impact of MLHC research. By anchoring our efforts in these principles, we transition from myopic technical novelty to a more balanced pursuit of solutions that are patient-centered, equitable, and ultimately transformative for healthcare.

### Selecting meaningful tasks

Given the field's hunger for technical novelty, MLHC research can sometimes focus on "solutions in search of problems" [83]. Researchers may regard healthcare primarily as a source of complex interesting data with compelling real-world justifications. There can be a tendency to focus on implementing the most interesting and novel techniques on healthcare data, rather than solving the most urgent and tangible healthcare problems. While "machine learning on healthcare data" can be a meaningful way to drive forward ML research, it must not be confused with true "machine learning for healthcare". Problems in healthcare are generally complex and context-specific.

We recommend:

- Distinguish between "machine learning on healthcare data" and "machine learning for healthcare problems", recognizing that the technical novelty of the former does not innately translate to impact on the latter.

- Engage with clinical stakeholders early in the clinical process, and working from problem to solution with earnest evaluation of whether a particular method—or machine learning at all —is most appropriate to address the problem.

### Moving past the status quo

A common approach in MLHC is using clinical data to predict variables corresponding with a clinician's decision, such as the diagnostic label of an X-ray [84,85], or the decision to prescribe a medication [86]. Such approaches implicitly treat that clinical assessment as the ground truth, and in so doing establish contemporary clinician-level performance as a ceiling. This is

problematic given the well-documented fallibility of human clinicians [87], and the pervasive biases which further worsen the quality of care for underserved patients [88,89].

While learning from clinicians (particularly ensembles of clinicians) is important, MLHC should strive where possible to surpass the quality and accuracy of existing clinical practice. A particularly illustrative example was provided by Pierson et al. [90], who demonstrated that training a knee X-ray algorithm to predict *patient-reported pain* rather than *radiologist-assessed arthritis burden* produced a model which eliminated pre-existing racial disparities in X-ray assessment.

We recommend:

- Focus wherever possible on prediction of either objective ground truth metrics such as mortality or relevant patient centered metrics such as reported pain, rather than strict agreement with human clinician assessments alone.

- Engage in ambitious attempts to elevate existing clinical standards, rather than simply striving to meet the baseline status quo.

### Establishing consistent responsibility standards

Current MLHC development lifecycles consider "performance improvement" in a given core metric such as area under the precision-recall curve (AUPRC) or accuracy metrics as the primary target [91–93]. The field is focused upon defining and carrying out narrow tasks, with the broader context and ultimate impact of the model often considered an afterthought [94]. This reality shapes every step of the MLHC process, from data collection, to model development, to validation—with issues of responsible AI such as fairness, accountability, and transparency considered after the fact if at all. While significant literature explores standards related to these issues [72,95–97], they are implemented in a patchy and inconsistent manner which renders comparison and accountability between institutions difficult.

We recommend:

- Establish and propagate industry standards for responsible AI development, incorporating patient-centric and ethical values. [83]

- Dethrone narrow performance metrics as the primary assessment method, and consider them alongside clinical impact and responsible AI issues throughout the development process.

### Operationalizing fairness

Fairness and equity are often posited as goals of MLHC development, however the terms are often used vaguely and without clear meaning, as previous criticism has noted [98]. Failure on the part of both the MLHC community and the broader regulatory apparatus to develop a clear set of expectations in this regard enables research with a wide range of actual impact to be passed off as sufficiently "fair". Far too often, the equity-focused analyses performed are only cursory, hidden away in supplemental tables and hardly engaged within the main body of a paper. Regardless of the specific consensus definition of the field, if fairness is to be achieved in MLHC, it must be baked into the process rather than painted on after the fact. It must be understood in concrete terms, and have the power to meaningfully shape the development process.

We recommend:

- Pursue consensus definitions of fairness in MLHC contexts, both procedural as well as outcome-based.

- Engage in subgroup analysis of the outputs of all MLHC projects, in order to understand and engage in dialogue and remediation regarding relevant disparities.

- Develop and hold projects to explicit standards regarding fairness, equity, and maximally permissible performance disparities between demographic groups.

- Render fairness constraints an integral part of training processes, rather than an ad-hoc or after-the-fact correction.

## Evaluation: Re-aligning incentives

Recognizing that innovation isn't merely about technical novelty but also about potential clinical impact can lead to a paradigm shift in how research is conducted and valued. The lure of common metrics of success can lead researchers to overlook the complex realities of real-world clinical contexts, and the sometimes subtle but significant disparities that exist between controlled research environments and the day-to-day chaos of healthcare provision. Whether through completing moonshot projects or solving overlooked last-mile challenges, MLHC researchers must work to create health equity, rather than merely managing bias. We advocate for a broadened perspective on the value of innovation, a more nuanced understanding of real-world applicability, and an appreciation for audacious endeavors, thus forging a path towards a more impactful MLHC landscape.

### Recognizing advances for more than just algorithmic/ technical novelty

Innovation is often equated with novelty in academic research, particularly in the increasingly intense competition for conference reviewer attention and approval. Commending research primarily for its originality encourages new and creative methods which may be ultimately only incremental in impact. At the same time, this philosophy acts to relatively disincentivize the unglamorous work necessary to develop an idea fully and address all of the various challenges which arise in its deployment in the clinical context. The result is an excess of incomplete research agendas, as researchers face rapidly diminishing returns for the additional follow-up work required to fully develop the implications of an initially novel work.

This issue is not merely in the realm of academic recognition, as both governmental grants and industry funding tend to follow a similar pattern. Given the increasing costs of model training, assessments of what will be fundable act to shape and ultimately bias research in directions which may be more attention-grabbing than impactful. Further, projects which are funded for technical novelty often lack access to the necessary follow-on funding to be maintained through the full life cycle and achieve impact.

We recommend:

- Recognize that novelty has multiple aspects by creating explicit research tracks highlighting technical significance and clinical relevance.

- Provide opportunities for follow-on funding to carry ideas forward to implementation and tangible impact.

- Conduct retrospective analyses of trends in machine learning applications for health [99], and introspect on areas that might require more research focus.

## Moving on from clinically unimportant metrics

MLHC research tends to focus on label prediction accuracy metrics such as sensitivity, specificity, and the precision-recall curve [91–93]. However, accuracy when predicting labels on a curated dataset does not always translate to accuracy in real-world clinical settings when deployed, often due to dataset shift [100,101] caused by critical and inescapable differences between the populations used in testing versus deployment [102]. More substantially, all accuracy metrics should be considered with respect to a clinical goal and a clinical workflow. Even an accurate prediction may have little clinical value [103] in certain circumstances—it may be too late, too early, about an event that cannot be prevented, or add little useful information to existing clinical assessments. Thus, the value of accuracy metrics as proxies for clinical relevance is highly variable and ultimately task- and context-specific.

We recommend:

- Develop and validate metrics in line with clinical goals and workflows, with an awareness of when information will augment meaningfully what clinicians already know [104].

- Validate clinical targets and metrics via clinical trials focused on tangible patient-relevant endpoints such as mortality or disability reduction.

- Pursue causal modeling for automated assessment of whether a given alert is likely to impact clinical decision-making.

- Evaluate machine learning solutions against already existing methods used in relevant healthcare settings as well as simple non-machine learning baselines.

- Monitor performance after deployment to identify the presence of dataset shift and help inform decisions about when model re-training is needed.

- Evaluate models in the local context of their intended deployment, recognizing that performance can be sensitive to subtle specifics.

## Encouraging moonshots

Stepping outside the comfortable bounds of topping established benchmarks purely *in silico* exposes a project to a wide range of uncertainties, and the most potentially impactful MLHC projects are often also the riskiest. They require seeking the approval of a wide range of stakeholders, and they often play out across extended multi-year timelines [105]. This can conflict with the career incentives of MLHC academics, particularly those on short tenure timelines, as well as students with brief PhD or even briefer MSc timelines. Researchers may fear—and with some justification—that pursuing such projects places the eggs of their career into a single fragile basket. If the field is going to succeed in having impact, this dynamic must be reversed.

We recommend:

- Recognize the insights gained from ambitious projects, even (or especially) if they ultimately are not successful.

- Encourage publication of process descriptions and intermediate results [106] for prolonged, multi-stage development and implementation.

- Allocate grant funding to ambitious projects with longer timelines, aiming at real-world clinical benchmarks and equitable outcomes.

- Publish "lessons learned" from projects that may not have succeeded at their original goals but nevertheless have insights of value for the rest of the field.

## Deployment: From code to clinic

The path from theory to practice is paved with often-overlooked aspects, including rigorous engineering, a profound understanding of human processes, acknowledgment of system limitations, anticipation of ongoing impact, and a commitment to parallel valuation of validation and innovation [105]. Each of these dimensions bears a unique set of considerations and requires explicit focus to ensure successful, safe, and sustainable implementation of MLHC. In the subsequent section, we delve into these complexities and offer recommendations that aim to foster an approach that is not just technically sound, but also holistic, human-centered, transparent, and future-oriented. Through these concerted efforts, we aspire to enhance the reliability, efficacy, and ultimately, the patient outcomes of MLHC deployment in diverse healthcare environments.

### Valuing engineering effort

Machine learning models require significant work to be deployed in practice, especially in high-stakes domains such as healthcare [107,108]. The scientific experiments accompanying the proposal of a new model, while necessary, are not sufficient to deem a model useful and practical [14]. Deployment involves significant work [107,108], such as cross-contextual validation, development of platforms or APIs for model access, online model monitoring, and data collection for further model improvements. All this accompanying work may, unfortunately, not be recognized as part of the research process, despite the fact that it is mandatory to achieve wide model use. This work is often separated out as "engineering" and deemed uninteresting in the academic context despite its presence as an essential component of any translation effort. This is particularly problematic in healthcare, given the variability and complexity of both the digital systems and sociotechnical structures involved.

The MLHC community should learn from and build upon preceding efforts in public health and health promotion in this regard. The Reach, Effectiveness, Adoption, Implementation, and Maintenance (RE-AIM) framework [109] is one robust example, which seeks to find a balance between the internal and external validity of projects by highlighting these challenges from the outset of a project. In particular, the MLHC community must come to regard the effort necessary to adapt to existing, complex healthcare contexts as core to, rather than ancillary to, any new project.

We recommend:

- Develop research tracks encouraging "last-mile" work, with an emphasis on deployment and impact evaluation.

- Create specific conference venues dedicated to engineering problems in machine learning for healthcare.

- Seek active collaboration with colleagues with expertise in data engineering, software engineering, systems engineering, bioinformatics, and related subfields.

- Develop and adopt standards that reduce the technical burden of organizations to deploy and monitor algorithmic performance in the real world setting.

- Utilize frameworks such as RE-AIM [109] to conceptualize and prepare for the work which must be done to adapt MLHC initiatives to real-world contexts.

### Considering human processes

While many MLHC projects are designed and reported in controlled computational isolation, all clinical processes involve the variability and complexity inherent to human involvement

[110]. Numerous questions arise at this juncture. How will model outputs be understood by clinicians? What does it take for a model's output to be trusted? Will the presence of an inaccurate model recommendation cause clinicians to second-guess their better judgment [111]? The MLHC field must work to further develop the multidisciplinary work that has been done to characterize human-machine systems, and the habits that clinicians build when working with machines. Well-characterized issues include automation bias [112–114] (overreliance on machine recommendations), algorithm aversion [115,116] (underreliance on machine recommendations), and alarm fatigue [117] (becoming overwhelmed by the frequency of alerts). Any approach which understands an ML model in isolation is fundamentally incomplete, and models developed in such isolation will fail to have tangible clinical impact.

We recommend:

- Work to understand the role of algorithms amidst the broader human-machine system [118].

- Engage clinicians, nurses, patients, and other end-users of systems in order to understand how model outputs are understood, and how their presentation can be optimized.

- Collaborate with colleagues in fields such as medical anthropology, human factors engineering, psychology, sociology, feminist techno-science, and user experience design.

- Establish processes for ongoing monitoring of clinician feedback, ensuring that the practical usability of models is continuously optimized.

- Ensure that appropriate backup procedures are in place to recognize and respond to model downtime or performance deterioration.

- Continue training clinicians for model-free circumstances, in order to avoid deskilling [119].

### Identifying system limitations

Current MLHC research is biased toward demonstration of the unique and novel capabilities of a model at its best. Yet, of equal importance is a robust understanding of what a given model *cannot* do. Characterizing and elaborating upon the particular failure cases of a model is essential to safe and impactful deployment, and enables amelioration measures such as learning when to defer to human clinician assessments [120,121]. Identification of limitations is also essential to ensure that a model is not uncritically deployed outside of the initial scope for which it was developed and validated.

We recommend:

- Reward researchers in peer review and publication processes for honest and thoughtful characterization of the limitations of a given model architecture.

- Ensure that regulatory processes require clear establishment of model limitations, and that these are consistently made clear to health systems and model end users.

- Work to generate standardized adversarial assessments [122] to probe the vulnerabilities of a given clinical model.

- Utilize standardized model reporting techniques, such as "Model Cards" [82] in order to ensure that model training contexts, targeted uses, and limitations are clearly disclosed.

## Building for continued impact

Rewards in academic MLHC research are heavily skewed toward the initial design and deployment of a model [100], whereas the ultimate clinical impact accumulates gradually over an extended period of time. The continued maintenance and execution of an MLHC project is a complicated endeavor in its own right, with issues such as dataset shift impacting model accuracy, and advancements in treatment protocols possibly impacting a project's overall appropriateness. Anticipating and addressing such factors must be considered an integral part of the MLHC enterprise, rather than being relegated as an unimportant ongoing maintenance task for the end-user [123]. Projects which are initiated but not maintained may even act as a net harm to patients overall, as processes may be altered and resources may be re-allocated in ways that are not easily reversed.

We recommend:

- Encourage the publication and dissemination of longitudinal assessments reporting retrospectively on the impact of models years after initial deployment.

- Establish guidelines and checklists to update and/or retrain models [123] in response to changing data and process environments.

- Establish institutional monitoring teams and processes for the longitudinal assessment and adjustment of clinical models for issues such as dataset shift.

- Offer rewards for long-lived, successful deployments of MLHC projects, as well as terminating projects assessed to have harms or negative impacts.

## Valuing generalization on par with innovation

Even where MLHC projects *are* successfully implemented at a single site, machine learning models experience a notorious degree of difficulty generalizing outside of their initial contexts. If MLHC is going to have an impact on patients broadly, there is significant work to be done in ensuring that the proverbial wheel does not require re-inventing in every hospital and clinic around the world [124]. At the other end of the spectrum, some models have been widely deployed without any significant effort toward validation, with deleterious consequences [124]. Yet as has been described with the replication crisis in psychology and other sciences, academic research conveys relatively little reward for the co-equal work necessary for validation.

Further, validation remains often project and context-specific, with a paucity of generalizable methods for comparison between projects. When considered in terms of patient impact, however, performance at each additional site is equally important to performance at the index institution. Methods such as "realist evaluation" [125] are highly useful in this regard, providing for formal structures to evaluate "what works, for whom, and in what circumstances" [126]. The framework ensures that contextual factors (such as the individuals involved, and their broader infrastructure) and context-specific mechanisms are explicit targets of evaluation and assessment.

We recommend:

- Dedicate increased focus to the techniques of transfer learning and local fine-tuning when necessary.

- Develop consistent interoperability standards, and methods for understanding the shifts in model performance as adapted to different contacts.

- Encourage publication of validation studies, with particular description of pitfalls faced in local translation.

- Create pipelines for the systematic comparison and validation of methods [80].

## Impact challenges in machine learning for healthcare

We believe that MLHC must be oriented first and foremost toward its impact upon the health of individual patients and the community at large. In the same vein as Wagstaff (2012) [1], we seek to offer a series of impact challenges which highlight tangible, meaningful goals for progress in the field [127].

1. A model deployed in an American urban hospital recognizes high-risk prenatal patients, recommends interventions, and successfully reduces the Black-white maternal mortality disparity by > = 50%.

2. A model deployed and maintained at a single center to predict patient deterioration [128] continues to maintain > = 95% of its initial predictive performance and impact on mortality across all demographic groups over a 10-year timeframe.

3. Data pipelines and transfer learning methods are used to deploy a chest x-ray diagnostic model [129] with significant clinical impact across urban and rural hospitals in three countries in Sub-Saharan Africa.

4. A "second opinion" diagnostic language model [130] for common clinical ailments is actively consulted as a part of regular outpatient clinical workflow with a >90% physician satisfaction rate with answer usefulness across multiple health systems in diverse socioeconomic contexts.

The purpose of these challenges is not to be prescriptive, but rather to highlight examples of concrete end-goals which the broader MLHC enterprise must be in service of. The structural recommendations highlighted elsewhere in this paper aim to create the contexts and processes —the "where", "who" and "how"—necessary to work toward such end-goals. Crucially, these goals do not need to be achieved from beginning to end by a single group, and MLHC endeavors do not need to be limited to grand projects. Rather, they are meant to frame and guide the field as a whole—that is, even smaller projects should be conceived, understood, and implemented in terms of how they move towards ambitious goals with tangible clinical impact.

Seeking to achieve (1), for example, would require moving beyond the task of simply predicting patient outcomes, and toward building MLHC as an integral part of a broader complex system considering the social dimensions of health. It would require not only technical expertise, but also consideration of patient needs and clinical barriers in a specific socio-technical context. In doing so, it would necessitate the collaborative involvement of members of a racially, culturally, and disciplinarily diverse team, which requires the expansion of inclusive training pathways, and efforts to promote MLHC among those interested in maternal health. Further, it would require the upstream expansion of data pathways to gather data outside of traditional academic centers, and the creation of open datasets necessary for fundamental ML methods to be applied to the field of maternal health.

Both machine learning and healthcare are highly complicated topics in isolation, and MLHC is in many ways even more so. We believe that a healthy and thriving MLHC community and research apparatus is one that is able to consistently, aim toward and ultimately achieve goals such as those embodied by these challenges. It is incumbent upon all of us

working in this field to work toward building such an apparatus—working not only to demonstrate technical novelty on data gathered amidst the background of healthcare, but to consistently generate solutions which work for all.

## Conclusion

Despite major advancements in the underlying technology, we contend that the impact of MLHC on meaningful clinical problems has been limited. This limitation in part reflects the inherent difficulty of deploying ML solutions in such a high-complexity and high-stakes domain. However, it also reflects significant issues in the structure and culture of the field of MLHC at large.

Data access [107,131] is woefully limited and inadequate to reflect the diversity of patient populations [132]. The research process often fails to select and engage with meaningful tasks. Evaluation tends to prize novelty over impact [1], and the field gravitates toward simplistic metrics. A majority of research and development efforts focus on technical "novel" model development rather than data collection and deployment. Underpinning this all, significant issues of bias in both data and process lead to the production of models which may only exacerbate existing healthcare disparities [27].

As we have discussed throughout this paper, these problems are real and substantial, but far from intractable. We are hopeful that this article may help to spark reflection and conversation within the MLHC field. We must work to distinguish between "machine learning research performed on healthcare data" and "machine learning for healthcare". We must re-orient ourselves toward the grand challenges which exist, and the tangible steps required to address them. We must strive to be a field defined first and foremost by our impact on the lives of patients and their families—that is, to engage in machine learning for healthcare that matters.

## Author Contributions

**Conceptualization:** Ioana Baldini, Leo Anthony Celi, Tristan Naumann, Uri Shalit.

**Investigation:** Aparna Balagopalan, Ioana Baldini, Leo Anthony Celi, Judy Gichoya, Liam G. McCoy, Tristan Naumann, Uri Shalit, Mihaela van der Schaar, Kiri L. Wagstaff.

**Methodology:** Aparna Balagopalan, Ioana Baldini, Leo Anthony Celi, Judy Gichoya, Liam G. McCoy, Tristan Naumann, Uri Shalit, Kiri L. Wagstaff.

**Supervision:** Ioana Baldini, Leo Anthony Celi, Judy Gichoya, Tristan Naumann, Uri Shalit, Mihaela van der Schaar, Kiri L. Wagstaff.

**Visualization:** Aparna Balagopalan.

**Writing – original draft:** Aparna Balagopalan, Ioana Baldini, Leo Anthony Celi, Judy Gichoya, Liam G. McCoy, Tristan Naumann, Uri Shalit, Mihaela van der Schaar, Kiri L. Wagstaff.

**Writing – review & editing:** Aparna Balagopalan, Ioana Baldini, Leo Anthony Celi, Judy Gichoya, Liam G. McCoy, Tristan Naumann, Uri Shalit, Mihaela van der Schaar, Kiri L. Wagstaff.

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
