## [Decision Letter · Decision Letter 0]

12 Dec 2023

PDIG-D-23-00335

Machine Learning for Healthcare that Matters: Reorienting from Technical Novelty to Equitable Impact

PLOS Digital Health

Dear Dr. McCoy,

Thank you for submitting your manuscript to PLOS Digital Health. After careful consideration, we feel that it has merit but does not fully meet PLOS Digital Health's publication criteria as it currently stands. Therefore, we invite you to submit a revised version of the manuscript that addresses the points raised during the review process.

Please submit your revised manuscript within 30 days Jan 11 2024 11:59PM. If you will need more time than this to complete your revisions, please reply to this message or contact the journal office at digitalhealth@plos.org. Please include the following items when submitting your revised manuscript:

We look forward to receiving your revised manuscript.

Kind regards,

Omar Badawi, Pharm.D., MPH

Section Editor

PLOS Digital Health

Journal Requirements:

2. Please send a completed 'Competing Interests' statement, including any COIs declared by your co-authors. If you have no competing interests to declare, please state "The authors have declared that no competing interests exist". Otherwise please declare all competing interests beginning with twhe statement "I have read the journal's policy and the authors of this manuscript have the following competing interests:"

3. Please ensure that Funding Information and Financial Disclosure Statement are matched.

4. In the Funding Information you indicated that no funding was received. Please revise the Funding Information field to reflect funding received.

5. Please provide separate figure files in .tif or .eps format only and remove any figures embedded in your manuscript file. Please also ensure that all files are under our size limit of 10MB.

Additional Editor Comments (if provided):

The manuscript is well written and would make an important contribution to the field. Please address the comments and thoughtful suggestions provided by the reviewers and we look forward to reading the revision.

Reviewers' comments:

Reviewer's Responses to Questions

**Comments to the Author**

1. Does this manuscript meet PLOS Digital Health’s publication criteria? Is the manuscript technically sound, and do the data support the conclusions? The manuscript must describe methodologically and ethically rigorous research with conclusions that are appropriately drawn based on the data presented.

Reviewer #1: Yes

Reviewer #2: Yes

2. Has the statistical analysis been performed appropriately and rigorously?

Reviewer #1: N/A

Reviewer #2: N/A

3. Have the authors made all data underlying the findings in their manuscript fully available (please refer to the Data Availability Statement at the start of the manuscript PDF file)?

Reviewer #1: Yes

Reviewer #2: Yes

4. Is the manuscript presented in an intelligible fashion and written in standard English?

Reviewer #1: Yes

Reviewer #2: Yes

5. Review Comments to the Author

Reviewer #1: This is a very good article. My suggestions are few but important. All of them pertain to the last two sections of the framework.

1. Under "Evaluation: Re-Aligning Incentives", you advocate for an appreciation of audacious endeavors. What do you mean by audacious? On the one hand, you want to elevate the overlooked realm of implementation science (the "last-mile" work); on the other, you want to encourage bold ideas. Suggest you align both of these thoughts toward the real-world goal of equity, as demonstrated by your final 4 examples. Those examples (I think!) demonstrate what you mean by audacity. Just make that plain here. We want people to ask how they can create equity using the model -- which is very different than asking how they can manage the bias of the model. It's a question of goals rather than boundaries.

2. Same section, under "Recognizing Advances..." You mention that researchers compete for reviewer attention. You should be more explicit about grant money and industry funding. The political economy of science shapes the research, which biases the research (be audacious and say it!).

3. Under "Moving on From Clinically Unimportant Metrics," you should add a recommendation about local testing and validation. It could fold into the clinical trials rec, though it's not exactly the same.

4. Under "Encouraging Moonshots," lengthen this recommendation: "Allocate grant funding to ambitious projects with longer timelines, aiming at real-world clinical benchmarks and equitable outcomes."

5. Under "Considering Human Processes," you ask three very good questions about how clinicians will interact with a model. I suggest you broaden the frame: what sort of habits are clinicians forming in relation to ML models? Then name specific risks, like automation bias and alarm fatigue (i.e. the habit of blindly following or blindly ignoring).

6. Under the same heading, in your recommendations, include: implement processes where clinicians can provide constant feedback. Establish who is accountable for the model, so that clinician feedback goes somewhere, and there are processes in place for continuous improvement (as well as triggers and backup plans for model downtime -- very important for patient safety).

7. Under "Identifying System Limitations", you recommend that researchers should be rewarded for stating the limitations of a model. Please ground this. How should they be rewarded? By whom, and with what incentive?

8. Under "Building for Continued Impact," suggest you state that short-term rewards pose a risk to patients.

9. In your recommendations for the same section, you recommend establishing institutional monitoring boards. "Boards" is too high-level. Suggest "teams and processes" or something like that.

10. Under "Impact Challenges..." you provide four excellent examples. Suggest you stay with the equity focus on number 4, just by lengthening it: "...a >=90% physician satisfaction rate with answer usefulness across multiple health systems in diverse socioeconomic contexts."

Overall, a really sharp set of ideas. A companion piece should map them to the AI lifecycle to further ground how this would go in a stepwise fashion: problem definition, data engineering, model building, small-scale testing, deployment and monitoring.

Reviewer #2: This landscape review highlights the disparity between the promise of machine learning for healthcare and the current state of the field, which emphasizes technical novelty over real world impact. The authors distinction between machine learning ON healthcare data and machine learning FOR healthcare is well articulated as is the framework proposed and recommendations suggested, which aim to achieve substantively meaningful and equitable clinical outcomes. Ultimately, the paper serves as a comprehensive critique of current state of the field but goes beyond simply pointing out shortcomings and concludes with impact challenges that refocus machine learning efforts within healthcare toward more meaningful societal impacts. This paper is an important contribution to the broader conversation around AI/ML and health equity and will be essential reading for all stakeholders in this space. 

- Fostering Diverse Dataset Creation

o Consider citing Datasheets for datasets as an example of a framework for providing detailed descriptions of how training datasets were created and how they should be used.

o Do you have any recommendations for resource constrained institutions around developing datasets?

- Opening Data and Increasing Accessibility 

o What does it mean/look like to “ensure efforts toward open data are inclusive” across various settings?

- Building Foundational Tools for MLHC Success

o Consider also emphasizing tools and platforms for archiving and sharing files and data for rigor and reproducibility (i.e., Dataverse.org)

- Understanding Social Determinants and Data Biases 

o Given the overarching goal of striving for equity, consider moving this section to the beginning above Fostering Diverse Dataset Creation.

o Introduce the concept of data empathy, the context/story behind the data that give way to inherent biases in the data: https://www.liebertpub.com/doi/full/10.1089/big.2014.0026

- Improving Diversity among Researchers 

o Provide some specific examples of how to incorporate diverse stakeholders into the ML design, development, and deployment processes.

o Add recommendation about sustainable investments in capacity building at historically black and minority serving institutions, which disproportionately train underrepresented health and computer science trainees.

- Valuing Engineering Effort 

o Consider introducing/recommending a formal implementation/evaluation framework, like RE-AIM, to quantify engineering efforts and standardize reporting of “issues, dimensions, and steps in the design, dissemination, and implementation process that can either facilitate or impede success” of model deployments.

- Identifying System Limitations 

o Have you considered recommending “Model cards” as a tool to help characterize and ensure dissemination of limitations of models.

- Valuing Generalization On Par With Innovation 

o Introduce the Realist evaluation framework, which provides a formal methodology for probing “What works, for whom, in what respects, to what extent, in what contexts, and how?”

- Other

o Ensure to have periods as the end of every recommendation.

6. PLOS authors have the option to publish the peer review history of their article (what does this mean?). If published, this will include your full peer review and any attached files.

**Do you want your identity to be public for this peer review?** For information about this choice, including consent withdrawal, please see our Privacy Policy.

Reviewer #1: No

Reviewer #2: No

---

## [Decision Letter · Decision Letter 1]

18 Feb 2024

Machine Learning for Healthcare that Matters: Reorienting from Technical Novelty to Equitable Impact

PDIG-D-23-00335R1

Dear Dr. McCoy,

We are pleased to inform you that your manuscript 'Machine Learning for Healthcare that Matters: Reorienting from Technical Novelty to Equitable Impact' has been provisionally accepted for publication in PLOS Digital Health.

Best regards,

Omar Badawi, Pharm.D., MPH

Section Editor

PLOS Digital Health

Reviewer Comments (if any, and for reference):

Reviewer's Responses to Questions

**Comments to the Author**

1. If the authors have adequately addressed your comments raised in a previous round of review and you feel that this manuscript is now acceptable for publication, you may indicate that here to bypass the “Comments to the Author” section, enter your conflict of interest statement in the “Confidential to Editor” section, and submit your "Accept" recommendation.

Reviewer #1: All comments have been addressed

Reviewer #2: All comments have been addressed

2. Does this manuscript meet PLOS Digital Health’s publication criteria? Is the manuscript technically sound, and do the data support the conclusions? The manuscript must describe methodologically and ethically rigorous research with conclusions that are appropriately drawn based on the data presented.

Reviewer #1: Yes

Reviewer #2: Yes

3. Has the statistical analysis been performed appropriately and rigorously?

Reviewer #1: N/A

Reviewer #2: N/A

4. Have the authors made all data underlying the findings in their manuscript fully available (please refer to the Data Availability Statement at the start of the manuscript PDF file)?

Reviewer #1: Yes

Reviewer #2: Yes

5. Is the manuscript presented in an intelligible fashion and written in standard English?

Reviewer #1: Yes

Reviewer #2: Yes

6. Review Comments to the Author

Reviewer #1: Thanks for addressing this reviewer's comments. Look forward to seeing the article published.

Reviewer #2: The authors addressed minor comments adequately.

7. PLOS authors have the option to publish the peer review history of their article (what does this mean?). If published, this will include your full peer review and any attached files.

**Do you want your identity to be public for this peer review?** For information about this choice, including consent withdrawal, please see our Privacy Policy.

Reviewer #1: **Yes: **Matthew Elmore

Reviewer #2: No
